# Nationwide Survey Reveals High Prevalence of Non-Swimmers among Children with Congenital Heart Defects

**DOI:** 10.3390/children10060988

**Published:** 2023-05-31

**Authors:** Christian Apitz, Dominik Tobias, Paul Helm, Ulrike M. Bauer, Claudia Niessner, Jannos Siaplaouras

**Affiliations:** 1Division of Pediatric Cardiology, Children’s Hospital, University of Ulm, 89075 Ulm, Germany; 2National Register for Congenital Heart Defects, 13353 Berlin, Germany; 3Institute for Sport and Sport Science, Karlsruhe Institute for Technology, 76131 Karlsruhe, Germany; 4Praxis am Herz-Jesu-Krankenhaus, 36037 Fulda, Germany

**Keywords:** congenital heart defects, pediatric cardiology, sports, swimming, prevention

## Abstract

Background: Physical activity is important for children with congenital heart defects (CHD), not only for somatic health, but also for neurologic, emotional, and psychosocial development. Swimming is a popular endurance sport which is in general suitable for most children with CHD. Since we have previously shown that children with CHD are less frequently physically active than their healthy peers, we hypothesized that the prevalence of non-swimmers is higher in CHD patients than in healthy children. Methods: To obtain representative data, we performed a nationwide survey in collaboration with the German National Register of Congenital Heart Defects (NRCHD) and the Institute for Sport Sciences of the Karlsruhe Institute for Technology (KIT). The questionnaire included questions capturing the prevalence of swimming skills and the timing of swim learning and was part of the “Motorik-Modul” (MoMo) from the German Health Interview and Examination Survey for Children and Adolescents (KiGGS). A representative age-matched subset of 4569 participants of the MoMo wave two study served as a healthy control group. Results: From 894 CHD-patients (mean age of 12.5 ± 3.1 years), the proportion of non-swimmers in children with CHD was significantly higher (16% versus 4.3%; *p* < 0.001) compared to healthy children and was dependent on CHD severity: Children with complex CHD had an almost five-fold increased risk (20.4%) of being unable to swim, whereas in children with simple CHD, the ability to swim did not differ significantly from their healthy reference group (5.6% vs. 4.3% non-swimmers (*p* = not significant). Conclusions: According to our results, one in five patients with complex CHD are non-swimmers, a situation that is concerning in regard of motoric development, inclusion and integration, as well as prevention of drowning accidents. Implementation of swim learning interventions for children with CHD would be a reasonable approach.

## 1. Introduction

Physical exercise and sports are of paramount importance for children with congenital heart disease (CHD), not just for the acquisition of motoric skills, but also for their cognitive, emotional, and psychosocial development. Current recommendations for physical activity, recreational sports, and exercise training in pediatric patients with CHD advise “to comply with public health recommendations of daily participation in 60 min or more of moderate-to-vigorous physical activity that is developmentally appropriate and enjoyable and involves a variety of activities” [1]. Swimming is a popular endurance sport, which is usually suitable for most children with CHD because of its mainly dynamic characteristics.

However, it has previously shown that children with CHD only rarely follow the recommended levels of physical activity. Various causes can be attributed to this: Cautiousness or even overprotection of parents, caregivers, and sports teachers, and furthermore, misperception of physicians and health care professionals regarding the risks and benefits of physical activity [2,3,4,5,6,7,8,9]. These uncertainties might result in the avoidance of contact with various sports, not least swimming.

However, being unable to swim is not only concerning in regard to development and integration but can also have dramatic consequences. Drowning remains a leading cause of evitable death in children [10,11,12]. According to statistical data from the German Life Saving Association (Deutsche Lebens-Rettungs-Gesellschaft, DLRG), 355 people died by drowning in Germany in 2022, and 46 of those were children or adolescents [13]. In addition, there were more than 100 near-drowning accidents making inpatient treatment necessary. Among children who survive near-drowning, health sequelae occur in about 7.5% of cases. The main reason for drowning or near-drowning in children over four years is minimal swim training or even a complete lack of training [11,12]. Steps to prevent drowning, therefore, include teaching children to swim, complemented by education on water safety [11].

In a previous national survey study, we were able to show that children with CHD in Germany were less frequently physically active than their healthy peers [14]. Therefore, we hypothesized that the prevalence of non-swimmers might also be higher in CHD patients than in healthy children.

## 2. Methods

To obtain representative data, we performed a nationwide survey in collaboration with the German National Register of Congenital Heart Defects (NRCHD) and the Institute for Sport Sciences of the Karlsruhe Institute for Technology (KIT). The cross-sectional online survey was conducted from November to December 2021.

For patient recruitment, the database of the NRCHD was scanned for patients with an age range from 6 to 17 years on the date of launching the survey. Patients and their respective families were contacted by email and invited to participate in the study. The protocol was approved by the institutional ethical committee.

The questionnaire included questions capturing the prevalence of swimming skills and the timing of swim learning and was part of the “Motorik-Modul” (MoMo) from the German Health Interview and Examination Survey for Children and Adolescents (KiGGS). Design and results of the MoMo Baseline and Longitudinal Study have been published previously [15]. A healthy control group served a representative age-matched subset of 4569 participants of the MoMo wave 2 study.

Statistical analysis was performed by IBM SPSS statistics version 25.0 (IBM Inc., Armonk, NY, USA). Values of continuous variables are reported as mean ± standard deviation. The Pearson’s chi-square test was used for group comparisons. A *p* value less than 0.05 was considered statistically significant.

## 3. Results

A total of 1647 patients agreed to participate in the online survey, and 894 patients (mean age of 12.5 ± 3.1 years; 47.2% female) completed the questionnaire. The study participants were allocated according to the anatomic complexity into simple (23.8%), moderate (37.8%), and complex CHD (38.4%) (Table 1) [16]. Genetic syndromes and chromosomal disorders were present in 72 patients (8%), most frequently Down syndrome in 45 patients (5%) and Di-George-Syndrome in 12 patients (1.3%) (Table 2).

Compared to healthy children, the prevalence of non-swimmers in children with CHD was significantly higher (16% versus 4.3%; *p* < 0.001) and was dependent on CHD-severity: Children with complex CHD had an almost five-fold increased risk (20.4%), children with moderate CHD a more than four-fold increased risk (18%) of being unable to swim, while in children with simple CHD, the ability to swim did not differ significantly from their healthy reference group (5.6% vs. 4.3% non-swimmers (*p* = not significant) (Figure 1).

Of those, who were specified as swimmers, the vast majority acquired their swimming skills between 4 to 12 years of age. Only 1.2% of CHD patients and 0.4% of healthy peers learnt swimming beyond the age of 12 years.

## 4. Discussion

Integration in the sense of inclusion, i.e., disabled people taking part in community life on equal terms, is a substantial component of the health care of children with chronic diseases including those with CHD. As sports activities are an integral part of successful integration, therefore, all patients with CHD can and should participate in physical activity and exercise. Restrictions should be avoided, except for patients with specific lesions or complications, who may require counselling regarding precautions and specific recommendations [1]. Swimming is a type of sport with positive effects on cardiorespiratory function and health, that can be recommended to most patients with CHD, and beyond being a competitive sport it is also an important leisure activity for social well-being.

However, according to this nationwide survey, more than 20 percent of patients with complex CHD are non-swimmers, thus not taking part in this important inclusion activity, and on the other side theoretically carrying the risk of drowning accidents.

There are obviously various potential reasons for this concerning situation. Usually, the swimming learning starts at the age of 4–6 years. However, this is an age, where patients with complex CHD such as Fontan patients frequently get involved to major surgeries or catheter-based interventions and therefore, especially at this point of life, things such as swim-learning must commonly be postponed, or even will be skipped completely. Concomitant handicaps due to syndromal disorders or neurological sequelae of previous interventions may also affect swim education in some patients. Additionally, misperceptions regarding the risks of swimming, as well as the overprotection of parents and physicians might also be potential contributing factors [8,9,14].

Recently, further obstacles arose due to the current global energy crisis with resulting increased energy costs more and more public indoor swimming pools have been closed or have markedly increased entrance fees, reducing the availability of access to swimming waters for children and adolescents. In addition, the lack of staff partly due to the COVID-19 pandemic results in a reduction of swimming lessons in elementary and secondary schools.

As an exemplary approach to providing relief, in 2022, the German Federal Association for children with CHD (Bundesverband Herzkranke Kinder e.V., BVHK) offered children with CHD family weekends with swimming lessons especially for children with CHD at various locations in Germany [17]. Most of these children are particularly dependent on very individual, competent support because every heart defect is different and every child with CHD has different limitations [1,18]. This is a very pragmatic approach and might improve inclusion and integration, as the acquisition of swimming skills might help to address the imbalance in opportunities available to children with a disability to participate in sports and physical activity with their healthy peers, in addition, might reduce or even avoid drowning in this sensitive patient group in the future.

The presented study has some limitations: Editing of the questionnaire was performed by the patients/families themselves. Therefore, data can theoretically be prone to bias, including recall bias and misjudgement as swimming skills might be over- or underestimated. Furthermore, the results might be affected by the social, educational, and economic situation of the children and their families.

The strength of this study, however, is the large-scaled sample size and the representative reference group.

## 5. Conclusions

According to our results, one in five complex CHD patients are non-swimmers, a situation that is concerning in regard to inclusion and integration, as well as the prevention of drowning accidents. Implementation of swim learning interventions for children with CHD would be a reasonable approach.

## Figures and Tables

**Figure 1 children-10-00988-f001:**
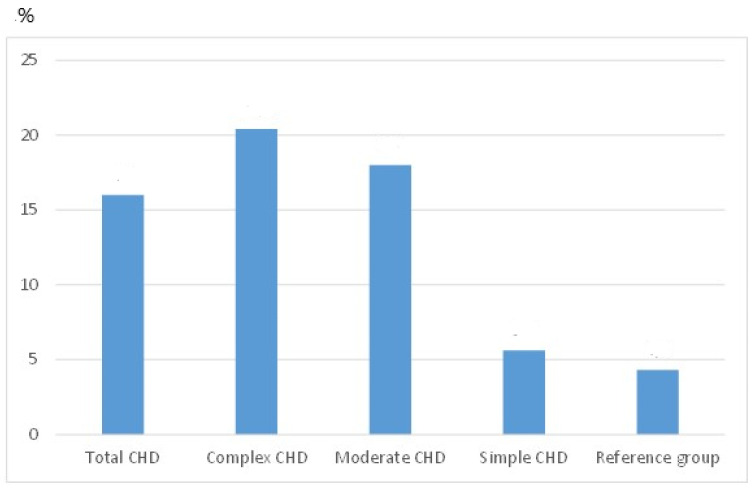
Prevalence of non-swimmers in children with congenital heart defects (CHD) and in the healthy reference group (in percent). Compared to healthy children, the proportion of non-swimmers in children with CHD was significantly higher (16% versus 4.3%; *p* < 0.001) and was dependent on CHD severity: Children with complex CHD had an almost five-fold increased risk (20.4%), children with moderate CHD a more than four-fold increased risk (18%) of being unable to swim, while in children with simple CHD, the ability to swim did not differ significantly from their healthy reference group (5.6% vs. 4.3% non-swimmers (*p* = not significant).

**Table 1 children-10-00988-t001:** Classification of congenital heart defects (CHD) of Task Force 1 of the 32nd Bethesda Conference categorizing according to the anatomic complexity into mild, moderate, and complex CHD (adapted with permission from Ref. [16], from Copyright Clearance Centers Rightslink service (Date 30 May 2023; No 5558971108981), Licensee: Christian Apitz).

Simple CHD	Moderate CHD	Complex CHD
Isolated congenital aortic valve disease	Aorto-left ventricular fistulas	Conduits, valved or nonvalved
Isolated congenital mitral valve disease (e.g., except parachute valve, cleft leaflet)	Anomalous pulmonary venous drainage, partial or total	Cyanotic congenital heart (all forms)
Small atrial septal defect	Atrioventricular septal defects (partial or complete)	Double-outlet ventricle
Isolated small ventricular septal defect (no associated lesions)	Coarctation of the aorta	Eisenmenger syndrome
Mild pulmonary stenosis	Ebstein’s anomaly	Fontan procedure
Small patent ductus arteriosus	Infundibular right ventricular outflow obstruction of significance	Mitral atresia
Previously ligated or occluded ductus arteriosus	Ostium primum atrial septal defect	Single ventricle (also called double inlet or outlet, common, or primitive)
Repaired secundum or sinus venosus atrial septal defect without residua	Patent ductus arteriosus (not closed)	Pulmonary atresia (all forms)
Repaired ventricular septal defect without residua	Pulmonary valve regurgitation (moderate to severe)	Pulmonary vascular obstructive disease
	Pulmonary valve stenosis (moderate to severe)	Transposition of the great arteries
	Sinus of Valsalva fistula/aneurysm	Tricuspid atresia
	Sinus venosus atrial septal defect	Truncus arteriosus/hemitruncus
	Subvalvular AS or SupraAS (except HOCM)	Other abnormalities of atrioventricular or ventriculoarterial connection not included above (ie, crisscross heart, isomerism, heterotaxy syndromes, ventricular inversion)
	Tetralogy of Fallot	
	Ventricular septal defect with: Absent valve or valves, Aortic regurgitation, Coarctation of the aorta, Mitral disease, Right ventricular outflow tract obstruction, Straddling tricuspid/mitral valve, Subaortic stenosis	

**Table 2 children-10-00988-t002:** Prevalence of Syndromes among the included CHD-patients.

Syndrome	Numbers of Patients	Percentage of Total (*n* = 894)
Down-Syndrome	45	5%
Di George-Syndrome	12	1.3%
Noonan-Syndrome	3	0.3%
Williams-Beuren-Syndrome	2	0.2%
Vacterl-Syndrome	2	0.2%
Edwards-Syndrome	2	0.2%
Charge-Syndrome	1	0.1%
Kartagener-Syndrome	1	0.1%
Holt-Oram-Syndrome	1	0.1%
XXY-Klinefelter-Syndrome	1	0.1%
Goldenhar-Syndrome	1	0.1%
Leopard-Syndrome	1	0.1%
Total	72	8%

## Data Availability

Additional data can be obtained from the authors.

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
