# Peer review of "Nationwide Survey Reveals High Prevalence of Non-Swimmers among Children with Congenital Heart Defects"

_children, 2023, doi:10.3390/children10060988_

Round 1

Reviewer 1 Report

Dear Authors,

We reviewed your manuscript on how to reduce the risk of drowning in children with congenital heart defects through swimming education. Topic is important, but I must say that the manuscript lacks clarity and depth in several areas.

Although previous studies have highlighted the positive effects of swimming as an exercise for cardiorespiratory function, health and life support, this study did not fully address the potential risks and limitations of children with congenital heart abnormalities. Moreover, its findings and interpretations are rudimentary and lack sufficient detail to be valuable to the sports and health sciences.

As reviewers for this paper, the authors suggest reconsidering the scope of the study and providing a more thorough analysis of the challenges and opportunities for swimming training in this particular population. The manuscript will also benefit from a clearer justification for the importance of this study in advancing our understanding of sports and health sciences.

Overall, I cannot recommend the publication of this manuscript in its current form. Authors are encouraged to address these issues and resubmit the corrected version for further review.

Best regards,

Author Response

Reviewer #1 Comments: 

Comment Reviewer 1: We reviewed your manuscript on how to reduce the risk of drowning in children with congenital heart defects through swimming education. Topic is important, but I must say that the manuscript lacks clarity and depth in several areas.

Although previous studies have highlighted the positive effects of swimming as an exercise for cardiorespiratory function, health and life support, this study did not fully address the potential risks and limitations of children with congenital heart abnormalities. Moreover, its findings and interpretations are rudimentary and lack sufficient detail to be valuable to the sports and health sciences.

As reviewers for this paper, the authors suggest reconsidering the scope of the study and providing a more thorough analysis of the challenges and opportunities for swimming training in this particular population. The manuscript will also benefit from a clearer justification for the importance of this study in advancing our understanding of sports and health sciences.

Response to Reviewer 1: Thank you for the comments, however we are a bit surprised regarding the strong criticism. We suppose that a different background of the reviewer and a kind of misunderstanding might be the reason. We therefore revised the text and tried to clarify the importance of our study. From the viewpoint of pediatric cardiologists, who are the target audience of this article, the results will be very interesting. Those who provide health care to children with congenital heart defects know the obstacles these patients have to struggle in their everyday life. Integration is a fundamental right and must be the treatment goal for each pediatric patient. Integration can be achieved with sports activities like swimming, therefore learning to swim is an important approach for integration, and can also help to avoid drowning accidents.

Maybe the presented results including the high number of non-swimmers among children with CHD does not advance understanding of sports and health sciences, as the reviewer mentioned, but it is indeed an important and concerning information for pediatric cardiologists, as it reflects inadequate inclusion.

Reviewer 2 Report

In the proposed manuscript, the authors tested the hypothesis that among children (6-17-year-olds) suffering from congenital heart defects (CHD) in Germany, compared to healthy children, there is a greater number of non-swimmers. The authors obtained contact information from the national register of persons with congenital heart defects, contacted the subjects and sent them a questionnaire via e-mail. The responses collected from 894 CHD patients were compared with age- and gender-matched data of 4,569 participants from the MoMo study. The comparison showed that the proportion of non-swimmers is significantly higher in the group of children with congenital heart disease than in healthy children, and that the proportion is higher the more complex the CHD.

The research question is clearly formulated, the sample size is adequate, the language and writing style are good, but the result is poor. In my opinion, regardless of the classification of this publication as "brief report", the presented results are not sufficient for publication. Is there a possibility for the authors to expand the study, present some other result? For example, to compare CHD swimmers and non-swimmers according to three categories of CHD classification you used, thus showing that it is realistic to expect that even children with complex CHD can learn to swim? Or maybe it turns out that no one with complex CHD can swim. Can authors determine how long it took people with CHD to learn to swim: the same, shorter or longer than healthy children?

Author Response

Reviewer #2 Comments: 

Comment Reviewer 2: The research question is clearly formulated, the sample size is adequate, the language and writing style are good, the result is poor. In my opinion, regardless of the classification of this publication as "brief report", the presented results are not sufficient for publication. Is there a possibility for the authors to expand the study, present some other result? For example, to compare CHD swimmers and non-swimmers according to three categories of CHD classification you used, thus showing that it is realistic to expect that even children with complex CHD can learn to swim? Or maybe it turns out that no one with complex CHD can swim. Can authors determine how long it took people with CHD to learn to swim: the same, shorter or longer than healthy children?

Response to Reviewer 2: Thank you for the comments, however we are a bit surprised regarding the strong criticism. We suppose that a different background of the reviewer and a kind of misunderstanding might be the reason. We therefore revised the text and tried to clarify the importance of the results of our study. From the viewpoint of pediatric cardiologists, who are the target audience of this article, the results will be very interesting. Those who provide health care to children with congenital heart defects know the obstacles these patients have to struggle in their everyday life. Integration is a fundamental right and must be the treatment goal for each pediatric patient. Integration can be achieved with sports activities like swimming, therefore learning to swim is an important approach for integration, and can also help to avoid drowning accidents.

Maybe the presented results including the high number of non-swimmers among children with CHD does not advance understanding of sports and health sciences, but it is indeed an important and concerning information for pediatric cardiologists, as it reflects inadequate inclusion.

Reviewer 3 Report

Dear authors,

Your submitted paper "Nationwide survey reveals high frequency of non-swimmers among children with congenital heart defects " could be beneficial for this field of science. However, I recommend some revisions.

Abstract:

Row 17 - National Register of Congenital Heart Defects (NRCHD). The official abbreviation is stated in https://www.kompetenznetz-ahf.de/en/about-us/register/ as NRAHF

 Introduction:

Row 36 - What do you mean, by stating the word "Frequency" of non-swimmers? The outcomes are used differences by percentage. I recommend using only the number of non-swimmers.

 Row 39 - 47 I think that you combine the introduction with part of the methods. You should state part of the methods as an individual paragraph after part of the introduction.

 Methods:

You should state separate parts of the methods, where it will be participants, the procedure for obtaining data, data treatment, and statistical analysis.

There is no clear how you obtained a representative set of healthy children.

There is no clear how you created individual groups "simple, moderate, and complex" You separated them according to the "guidelines for the management of adults with congenital heart disease"? as you stated in the references list under the number 7

You should explain better this important things in the part of methods

Results:

The graph should show the significant differences between groups and, in the notice, you should describe the abbreviation.

Discussion:

Discussion should be separated from other texts. You should answer the hypothesis that was stated in the part of the introduction.

Overall, I recommend properly stating the methods and explaining better some parts that I stated above and separating the results and discussion, and clearly interpreting obtained results in conclusion.

Even though the methodology of this paper is not clearly specified thus, the contribution and aim of this study are clear and beneficial for scientists as well as for practical use.

Author Response

Reviewer #3 Comments: 

3.1 Your submitted paper "Nationwide survey reveals high frequency of non-swimmers among children with congenital heart defects " could be beneficial for this field of science.

Response 3.1: We thank the reviewer for this positive comment.

3.2 Abstract: Row 17 - National Register of Congenital Heart Defects (NRCHD). The official abbreviation is stated in https://www.kompetenznetz-ahf.de/en/about-us/register/ as NRAHF

Response 3.2: Thanks for this comment. We double-checked this with the director of this institution: NRAHF is the German abbreviation, but the official international abbreviation is NRCHD (https://dzhk.de/forschung/klinische-forschung/alle-studien/studie/detail/nrchd/)

3.3 Introduction: Row 36 - What do you mean, by stating the word "Frequency" of non-swimmers? The outcomes are used differences by percentage. I recommend using only the number of non-swimmers.

Response 3.3: Thank you for this comment. We changed this accordingly.

3.4 Row 39 - 47 I think that you combine the introduction with part of the methods. You should state part of the methods as an individual paragraph after part of the introduction.

Response 3.4: Thank you for this comment. We changed this accordingly.

3.5 Methods: You should state separate parts of the methods, where it will be participants, the procedure for obtaining data, data treatment, and statistical analysis.

Response 3.5: Thank you for this comment. We changed this accordingly.

3.6 There is no clear how you obtained a representative set of healthy children.

Response 3.6: We compared our data with a representative age-matched subset of 4,569 participants of the ”Motorik-Modul“ (MoMo) wave 2 study from the German Health Interview and Examination Survey for Children and Adolescents (KiGGS), which was provided to us by the leading organization for this modul of the KiGGS study, the Institute for Sport and Sport Science, Institute for Technology, Karlsruhe, Germany.

3.7 There is no clear how you created individual groups "simple, moderate, and complex" You separated them according to the "guidelines for the management of adults with congenital heart disease"? as you stated in the references list under the number 7

Response 3.7: We have added table 1 with the Warnes-classification of CHD severity. This is a frequent used classification in order to allocate patients into the three CHD-severity groups.

3.8 You should explain better this important things in the part of methods

Response 3.8: Thank you for this comment. We changed this accordingly.

3.9  Results: The graph should show the significant differences between groups and, in the notice, you should describe the abbreviation.

Response 3.9: We have added a more detailed description including the significance level to the figure legend. The abbreviation is now described.

3.10 Discussion: Discussion should be separated from other texts. You should answer the hypothesis that was stated in the part of the introduction.

Response 3.10: Thank you for this comment. We changed this accordingly. The answer to the hypothesis is presented.

3.11 Overall, I recommend properly stating the methods and explaining better some parts that I stated above and separating the results and discussion, and clearly interpreting obtained results in conclusion.

Response 3.11: Thank you for this comment. We changed this accordingly.

3.12  Even though the methodology of this paper is not clearly specified thus, the contribution and aim of this study are clear and beneficial for scientists as well as for practical use.

Response 3.12: We thank the reviewer for this positive comment, and have now updated the methods section accordingly.

Round 2

Reviewer 1 Report

Although previous studies have highlighted the positive effects of swimming as an exercise for cardiorespiratory function, health and life support, this study did not fully address the potential risks and limitations of children with congenital heart abnormalities. Moreover, its findings and interpretations are rudimentary and lack sufficient detail to be valuable to the sports and health sciences.

Reviewer 2 Report

Despite the authors` explanation, I still think that descriptive statistics based on only one question answered in an online questionnaire, is a rather modest result of the study.

Reviewer 3 Report

Dear authors,

Thank you for rewriting and explaining some issues that I recommended to fix it. However, even though I agree that this article has a clear aim and practical use, thus the part of the methods is described a little bit weakly for the type of this study.